# Geographic Information Visualization and Sustainable Development of Low-Carbon Rural Slow Tourism under Artificial Intelligence

**Gongyi Jiang [1,2], Weijun Gao [2,3], Meng Xu [4,*], Mingjia Tong [1] and Zhonghui Liu [5]**

[1]  Foreign Languages Department, Tourism College of Zhejiang, Hangzhou 310043, China
[2]  Faculty of Environmental Engineering, The University of Kitakyushu, Kitakyushu 808-0135, Japan
[3]  Innovation Institute for Sustainable Maritime Architecture Research and Technology, Qingdao University of Technology, Qingdao 266033, China
[4]  Chemical Engineering and Technology, Zhejiang University of Technology, Hangzhou 310014, China
[5]  School of Municipal and Environmental Engineering, Jilin Jianzhu University, Changchun 130118, China
[*]  Correspondence: xumeng@zjut.edu.cn

**Abstract:** This study conducts in-depth research on geographic information visualization and the sustainable development of low-carbon rural slow tourism under artificial intelligence (AI) to analyze and discuss the visualization of geographic information and the sustainable development of low-carbon slow tourism in rural areas. First, the development options related to low-carbon tourism in rural areas are discussed. Then, a low-carbon rural slow tourism recommendation method based on AI and a low-carbon rural tourism scene recognition method based on Cross-Media Retrieval (CMR) data are proposed. Finally, the proposed scheme is tested. The test results show that the carbon dioxide emissions of one-day tourism projects account for less than 10% of the total tourism industry. From the proportion, it is found that air transport accounts for the largest proportion, more than 40%. With the development of time, the number of rural slow tourists in Guizhou has increased the most, while the number of rural slow tourists in Yunnan has increased to a lesser extent. In the K-means clustering model, the accuracy of scenario classification based on the semantic features of scene attributes is 5.26% higher than that of attribute likelihood vectors. On the Support Vector Machine classifier, the scene classification accuracy based on the semantic features of scene attributes is 19.2% higher than that of the scene classification based on attribute likelihood vector features. CMR techniques have also played a satisfying role in identifying rural tourism scenarios. They enable passengers to quickly identify tourist attractions to save preparation time and provide more flexible time for the tour process. The research results have made certain contributions to the sustainable development of low-carbon rural slow tourism.

**Keywords:** artificial intelligence; low-carbon villages; slow tourism; visualization; sustainable development; Cross-Media Retrieval technology; scenario recognition

## 1. Introduction

Global climate and environmental problems are becoming increasingly serious. In 2003, the British government issued the energy white paper "Our Energy Future: Creating Low Carbon Economy," which pioneered the concept of the "low-carbon economy." "Low-Carbon Tourism (LCT)" is the specific application of the concept of the "low-carbon economy" in the tourism industry. In 2009, the World Economic Forum released the report "Towards Low-Carbon Travel and Tourism," officially proposing the concept of "LCT." Its core is to reduce carbon dioxide emissions in the process of tourism, harmonize tourism development with the environment, and promote the sustainable development of tourism destinations. In recent years, with the support of national policies and the expansion of the leisure needs of urban residents, rural tourism has developed rapidly in China. The

total number of rural tourism trips increased from 720 million in 2012 to 2.82 billion in 2018, with an average annual compound growth rate of 31.2%. The influx of people, logistics, and information inevitably brings great challenges while bringing unlimited opportunities to the countryside. The ecological environment is fragile. The environmental protection system and mechanism are not perfect. The realization of environmental protection technology is difficult. The awareness of environmental protection lags. Thus, the phenomena of rural resource destruction, environmental pollution, and cultural conflicts are increasingly prominent. Rural tourism is a form of leisure for urban residents and a way for farmers to move away from poverty and become rich. It should also take more ecological responsibility. Therefore, the development of low-carbon rural tourism is the only way to transform and upgrade rural tourism. The low-carbon economy adjusts and improves the traditional economic development model and lifestyle through innovative systems and technologies to realize the development model of the ecological economy. Its main features are "three low and three high," which are low energy consumption, low pollution, low emissions, high efficacy, high efficiency, and high benefit. LCT refers to a new tourism development method that uses low-carbon economic theory as a guide to develop and utilize tourism resources and the environment to achieve high efficiency and low consumption and minimize environmental damage to resource utilization. It includes the decarbonization of tourism production and tourism consumption. Therefore, low-carbon rural tourism is a kind of deep-level environmental protection tourism that obtains the sustainable development of rural tourism under the guidance of low-carbon economic theory and LCT thinking in the operation of the rural tourism system. It also includes the decarbonization of rural tourism production and rural tourism consumption [1].

A Geographic Information System (GIS) is a specific spatial information system. It uses modern methods to collect, store, analyze, manage, display, and simulate spatial information systems related to the spatial distribution of geographic data. Its core technology is computer science and technology. The basic technology is the database, map visualization, and spatial analysis technology. The emergence of GIS provides the fastest, most convenient, and most accurate methods and means for complex data management, multisource achievement expression, and spatial data analysis in the tourism industry. Tourism GIS is a decision support system that describes the collection, storage, processing, analysis, and output of tourism information. It is based on the digitalization of tourism information and adopts geographic model analysis methods to provide various pieces of spatial and dynamic tourism information in real time. It is a computer technology system that provides decision-making management and services to society for tourism management departments. The tourism industry is comprehensive. Playing, traveling, eating, staying, entertainment, and shopping are the six elements of tourism services. Many industries can join the tourism industry. With the development and popularization of Internet technology, people's tourism concepts and methods have changed, which has promoted the further development of tourism. The existing level of informatization in the tourism industry is far from meeting the needs of industrial development. For example, tourist information products and technological means targeting tourists cannot meet the needs of the public. Its advocacy and services also rely heavily on traditional means [2]. Interaction with potential visitors cannot be provided. The information involved in tourism resources has obvious geo-reference characteristics and spatial analysis characteristics. GIS technology is the best technical platform to solve the contradictions and development bottlenecks encountered by the tourism industry at this stage. The tourism public information service platform based on Web GIS technology can not only meet the needs of tourists to the greatest extent but also effectively improve the tourism investment environment [3].

Traditionally, Cross-Media Retrieval (CMR) employs scenario-based semantic learning models for image data, such as the scenario recognition method based on the Fisher vector and the scenario recognition method based on target recognition [4]. Zhan et al. (2021) proposed the CMR-oriented representation method and scenario recognition method based on Deep Convolution Neural Network (DCNN). They used the shared pooling layer to transform the visual scenario features and text features into a shared semantic pooling layer. They achieved the goal of cross-media semantic space sharing. Meanwhile, the cross-media scenes were uniformly represented to construct the cross-media scenario recognition dataset [5]. Zhang et al. (2022) proposed a CMR knowledge transfer method based on Deep Learning (DL) [6]. Moghadam et al. (2021) designed a hybrid multimodal DL method mainly for short-term traffic flow prediction [7]. Additionally, Che et al. (2020) studied a relevant Recurrent Neural Network (RNN) model to solve the semantic learning problem of multimode spatiotemporal data in cross-media. They realized the semantic learning between video and audio data through the proposed model [8]. Nasralla (2021) proposed a new approach to guide the design and development of sustainable rehabilitation systems using time series analysis to identify faulty Internet of Things devices. Devices used Machine Learning (ML) on time series to identify and predict failures to support sustainability [9]. Chen et al. (2023) [10] proposed that carbon dioxide emissions are one of the important causes of climate problems such as global warming. Exploring the development and evolution trend of residential carbon emission research is of great significance for mitigating global climate change. However, there has not been a comprehensive review of research in this area of research. They used CiteSpace bibliometric analysis software to draw a visual knowledge map of residential carbon emission research, revealing its research status, research hotspots, and development trends. Dai et al. (2023) adopted a logical deduction method, derived the specific path of DIT to drive the low-carbon transformation of agriculture from the application status of Digital Imaging Technician (DIT), and further discussed the problems that may arise in this process and the corresponding solutions [11].

The existing literature has researched LCT but is mainly aimed at designing green tourism schemes, and the research objects are non-rural tourism sites. There has been little research on rural slow tourism and LCT. The main objectives of this study include the following aspects: (1) The first is to interpret the connotation of rural slow tourism and initially establish a research framework for rural slow tourism space. At this stage, there is no clear reference to "rural slow tourism." It is hoped that the concept of "rural slow tourism" can be defined accordingly by summarizing the domestic and foreign research on rural tourism and slow tourism. The concept, characteristics, and scope of the application of rural slow tourism can be clarified. Moreover, components of the rural slow tourism space can be extracted. Therefore, it can provide a theoretical basis for summarizing the spatial construction method of the specific case of rural slow tourism. (2) The second is to interpret the requirements of the rural revitalization strategy for rural tourism and clarify the starting point of spatial research. At present, in response to the rural revitalization strategy, various sports involving rural tourism and rural construction are emerging one after another. As a favorable policy for developing rural areas and realizing a beautiful living environment in rural areas, rural revitalization aims to achieve both rural protection and rural economic development to enhance the sense of identity and belonging of the vast number of farmers. However, economic development and the preservation of rural landscapes ultimately need to be implemented in space. Therefore, under the perspective of a clear rural revitalization strategy, the direction and focus of China's rural development is the starting point for the creation of a rural slow tourism space. Based on this, this study starts from the rural tourism scene and uses various ML methods to explore new ways to achieve LCT. Specifically, the research methods include literature analysis and theoretical model construction, and the combination of the two methods can ensure the reliability of the experimental results. Here, the geographic information visualization and sustainable development of rural slow LCT are analyzed and studied based on AI. The role and influence of DL combined with AI in rural slow LCT are discussed. The development

path, geographic information visualization, and sustainable development of rural slow LCT are deeply explored and studied. The data analysis and research are carried out using AI, which enriches the research results of the sustainable development of rural slow LCT in China. This study also uses CMR technology to propose the Scenario Attribute Semantics Feature (SASF) extraction algorithm for rural tourism scenario recognition. It sets up experiments to verify the recognition effect of the algorithm. The innovation of this study lies in the data analysis and research on the geographic information visualization and sustainable development of rural slow LCT through AI, which plays an important role in realizing rural slow LCT in China and promoting the sustainable development of rural slow LCT in China. Here, the current status of domestic and international research on rural slow tourism is analyzed by compiling and summarizing the literature. The concept of rural slow tourism is pioneered based on a full understanding of the relevant concepts. Combined with the results of theoretical research and the study of specific cases, the strategy of rural slow tourism space creation is proposed. Section 1 expounds on the development status of the tourism GIS, compares the traditional tourism information system and tourism GIS, and introduces the purpose and significance of the establishment of the tourism GIS. Section 2 analyzes the system requirements from the perspective of information asymmetry in the tourism market and elaborates on the key technologies for development. Section 3 is the system architecture and overall functional design. Section 4 contains the conclusions and recommendations.

The structural diagram of this study is displayed in Figure 1.

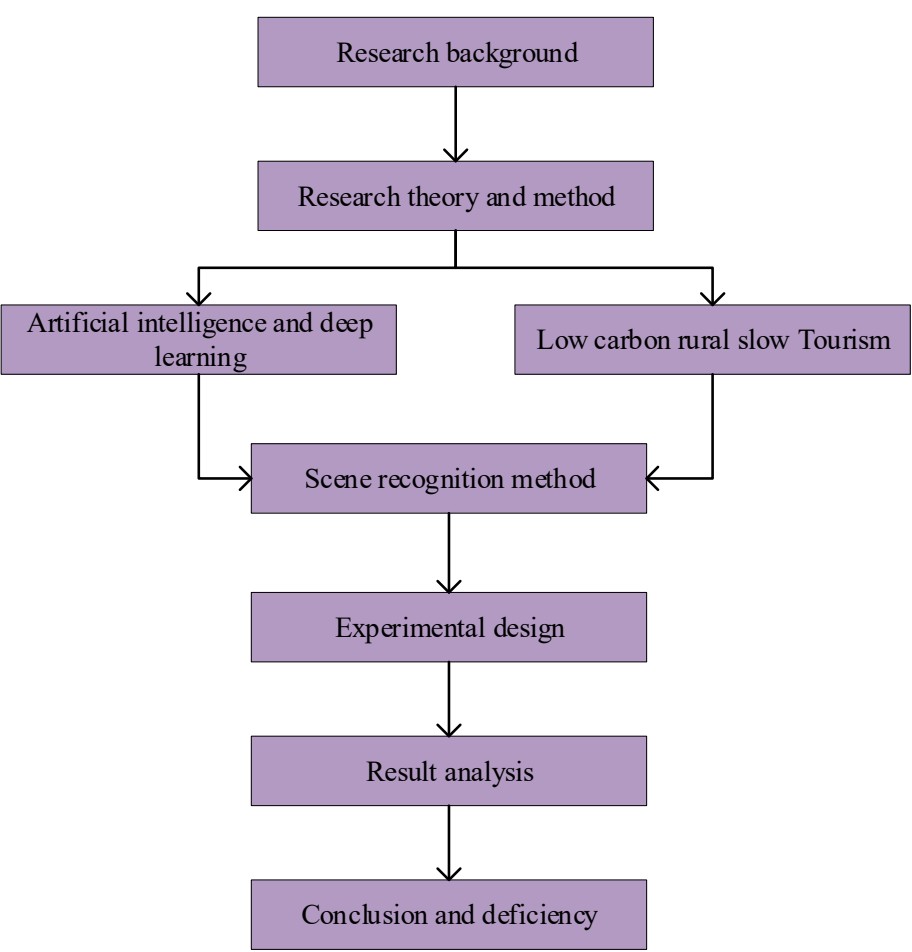

**Figure 1.** The structure of this study.

## 2. Related Concepts and Methods

### 2.1. Overview of the Relevant Theories of AI and DL

With the development of information technology, various sciences and technologies are developing rapidly, and they are linked. AI, one of the branches of computer science, evolved from the research and development of simulation and extension of human intelligence and the research focus of experts and scholars. It has developed into a new technology with great challenges and rich content. AI tries to reveal the essence of intelligence and produce an intelligent machine that can respond in a similar way to human intelligence. The research based on AI includes language recognition, image recognition, and Natural Language Processing (NLP), and AI has entered all aspects of society and people's lives with increasingly mature theories and technologies, such as machine vision, fingerprint recognition, face recognition, and intelligent search.

DL is a new research direction in ML, which is introduced to make ML closer to AI. It can explore the inherent law and the representation of sample data. The information obtained in the learning process can help to interpret data. Its ultimate goal is to enable machines to have the same analytical learning ability as humans and recognize texts, images, and sounds. As a complex ML algorithm, DL has superior research results in speech and image recognition.

### 2.2. Overview of the Theory of Rural Slow LCT

Low carbon means low gas emissions in greenhouses. Greenhouse gases are in the atmosphere, absorbing solar radiation reflected to the ground and re-emitting it. The low-carbon economy is an economic development model based on the existing environment. It refers to the economic development model under the concept of sustainable development through technological innovation, institutional innovation, industrial transformation, and new energy development to minimize high-carbon energy consumption, reduce greenhouse gas emissions, achieve economic development and the win-win of the ecological environment, and create ways and opportunities to achieve higher living standards and improve people's life quality [12]. A low-carbon economy is a concept, and it changes people's ways of life. Since it was introduced in China, the low-carbon economy has become an important prerequisite for sustainable development and also a vital means to achieve sustainable development. The current research results show that the low-carbon economy is an economic model based on low energy consumption, low emissions, and low pollution, and it is also revolutionizing the human development model [13].

Low-carbon slow tourism, borrowing the concept of a low-carbon economy, is a new type of tourism with low energy consumption and low pollution as the core. As a comprehensive industry, tourism involves many aspects such as the travel and entertainment of tourists. It responds to the concept of a low-carbon lifestyle, implements the carbon mechanism, and achieves low-carbon achievements. It is a frontier project to realize the development mode of a low-carbon economy [14]. The division of resources and energy consumption in a low-carbon economy is listed in Table 1 [15].

**Table 1.** Resource consumption project division of low-carbon tourism.

| Resource Consumption Project | Eating | Staying | Traveling | Shopping | Entertainment | Playing |
|---|---|---|---|---|---|---|
| Resources consumed by the project | Biology/ energy | Building/ energy | Building/ energy | Biology/ energy | Building/ energy | Building/ energy |

Sustainable development of tourism means that tourism development is required to meet the needs of tourists and residents and the possibility of tourism development to ensure the interests of future generations of mankind. In other words, the sustainable development of tourism refers to the sustainable and coordinated development of tourism and nature, society, culture, and living environment by considering the role and premise of tourism in the economic, social, cultural, and natural environment. The sustainable

development of slow rural tourism can make the development of slow rural tourism meet the spiritual needs and psychological needs of most people, promote agricultural development, improve the living standards of farmers, increase employment opportunities for the surplus labor, make contributions to the adjustment of rural industry, and promote the local economic development. As a result, a new countryside is constructed. Geographic information visualization of slow rural tourism is realized by graphics, computer graphics, image processing technology, the local geological information input, processing query, analysis, and prediction of the results of the data in the form of graphic symbols, icons, text, and video visualization and interactive theories, methods, and technologies [16,17].

Slow tourism is a modern lifestyle formed based on slow culture. It has the basic concepts of low-carbon travel, the pursuit of freedom, and slow travel time. People pursue personal experience in slow tourism. The concept of rural slow LCT is to excavate the rural life of slow tourism in the rural slow LCT planning, focus on the experience of contemporary rural life, and integrate the slow pace of life into slow rural tourism. This is the product of people's pursuit of a new tourism model due to the rapid development of urbanization. This tourism model is a rural slow LCT that taps resources from the rural interior, develops leisure and entertainment, and plays a significant role in the economic development of China's modern rural slow tourism [18].

### 2.3. Research Methods of AI and Low-Carbon Rural Slow Tourism

In the analysis and research of rural slow LCT, the commonly used methods include literature research, inductive analysis, quantitative analysis, and empirical research. First, the low-carbon and slow rural tourism and the theory of AI are studied through much relevant data in China and foreign countries, and the current situation of China's rural areas is summarized by the literature review. Second, carbon emissions are investigated and summarized in implementing slow rural tourism in China. The low-carbon measures are achieved in the target region, providing a reliable theoretical basis for future research. Finally, a place is taken as empirical research. The geographic information visualization and sustainable development of low-carbon rural slow tourism based on AI are summarized. Corresponding policy measures are put forward, facilitating the construction and development of low-carbon rural slow tourism in other regions of China [19].

The application of AI to rural slow LCT is the use of relevant models in data analysis. The main entities in the data analysis of the rural slow LCT model under AI are tourists (M) and rural slow LCT destinations (W). Rural slow LCT is developed from various rural slow tourism suburbs or villages with farmers and rural life as the theme throughout the country [20], and the location of rural slow LCT is associated with the specific POI (Point of Interest). In a GIS, a POI can be a house or shop, which is defined by a set of geographical coordinates.

For a rural slow LCT destination wi $\in$ W, its annotation can be expressed by tuples, as shown in Equation (1).

$$\xi_{w_i} = (\chi_1, \ldots, \chi_i; n_1 \ldots n_i) \tag{1}$$

In Equation (1), $\chi_i$ and $n_i$ are ontology properties and metadata, respectively. One or more affiliated documents of low-carbon rural tourism destinations (such as custom introductions related to rural characteristics, photos and videos of landscapes, and other related data) are associated with a given rural slow LCT destination.

Tourists' static information and dynamic information need to be recorded. Static information is recorded in the configuration file of tourists, and dynamic information is recorded in the data log of tourists. For tourist mi, the tourist configuration file can be represented by tuples, as demonstrated in Equation (2).

$$v_{m_i} = (a_1, \ldots, a_i; b_1 \ldots b_i) \tag{2}$$

In Equation (2), $a_i$ and $b_i$ are the metadata of the tourists' ontology properties and configuration files, respectively.

Given a set of rural slow LCT destinations W and a set of tourists M, the data log of tourists can be represented by tuple sets, as shown in Equation (3).

$$p = (T, m, w; \bar{x}_1, \ldots \bar{x}_n) \tag{3}$$

In Equation (3), $T$ is time, $m$ and $w$ are tourists and the destinations of rural slow LCT, respectively, and $\bar{x}_1, \ldots, \bar{x}_n$ is the behavior attribute of tourists to a certain object.

These data can form a complete database, in which the analysis of tourists' information data can infer what rural slow tourism projects tourists may prefer and create targeted rural slow LCT destinations according to these tourists' data [21].

### 2.4. Recommendation Methods of Low-Carbon Rural Slow Tourism Based on AI

When tourists search the rural slow LCT destinations, the system first learns the data information of tourists through in-depth learning. The social network selects a collection of rural slow LCT destinations that the tourists may be more interested in based on the location and interests obtained. Based on the dynamic analysis of tourists' social network behavior, the selected rural slow LCT destinations are divided into three different directions, namely tourists' interests, their emotions, and the popularity of the destinations. Finally, combined with the location of tourists and the attraction of the tourism destinations, time, and space, the most likely choice of rural slow LCT destinations is achieved [22].

In this process, tourists use DL to select rural slow tourism destinations and learn the tourist profile to the final subset D. Moreover, DL is used to calculate the probability of each destination relative to the tourist configuration file: the tourist destination with a threshold greater than a fixed y is added to subset D. The initialization subset D is an empty set. When wi ∈ W and y ≥ 0, the rural slow LCT destination is added to the subset D.

$$p\left(\frac{\alpha}{w}\right) = \frac{p\left(\frac{w}{\alpha}\right)p(\alpha)}{p(w)} \geq y \tag{4}$$

After the approximate rural slow LCT subset is obtained, DL is carried out according to the data of tourists, and the computer uses the adaptive algorithm to calculate the rural slow LCT destination with the highest score of tourists' interests and hobbies, the popularity of the destination, and NLP.

### 2.5. Low-Carbon Rural Tourism-Oriented Scenario Recognition Method Based on CMR Data

Currently, tourism generally cannot identify a tourism scene well to prepare for a comprehensive tourism strategy. Therefore, studying an efficient scenario recognition method can provide a good auxiliary role in choosing tourism sites.

#### 2.5.1. SASF Algorithm Description

The event-triggered concurrent data flow model is proposed based on dynamic pure data flow. Generally speaking, it can be divided into three levels. The first is the data input layer. Its main function is to obtain data from physical hardware or generate data on its own. It is the source of data for the data processing layer. The second is the data processing layer. It completes various types of data processing, such as filtering and control algorithm calculations. The third is the data output layer, which includes the output of data from the network. It outputs data to memory or display. From the perspective of real time, the priority of the data input layer should be higher than the priority of the data processing layer, and the priority of the data processing layer should be higher than the priority of the data output layer.

A set of tag words is extracted from the document describing a scenario to label the scenario. These tags are the attributes of the scenario. A set of attributes describes the semantics of a scenario, and a phrase can express each attribute. The semantic features are composed of the attribute semantics in the scenario and the semantic relationships between attributes. It is assumed that the scenarios are similar, and the corresponding attribute

semantics and semantic attribute relationships of these scenarios are also similar. These text words with attribute semantics cannot be directly applied to CMR computing. Thus, attribute semantic features must be extracted. The framework of the SASF is specified in Figure 2.

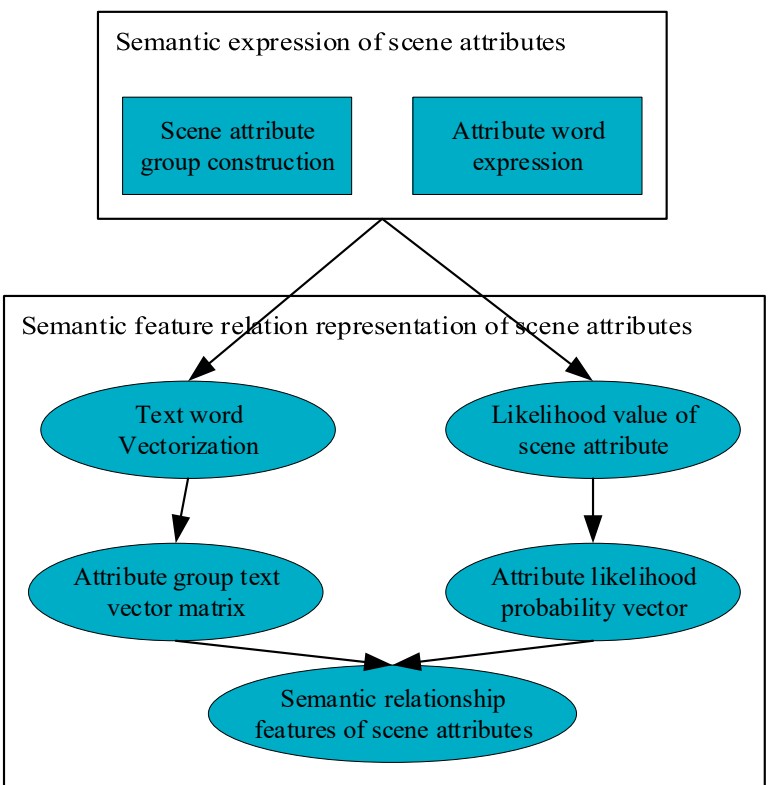

**Figure 2.** Schemes follow the same formatting.

2.5.2. Semantic Expression of Tourism Scenario Attributes

For the construction of the tourism scenario attribute space Am, word space Φn contains words that describe the travel scenario. According to the semantic meaning of words and the classification of scenario attributes, the word space Φn is mapped to attribute space Am and formalized as f: Φn→Am.

The word space of the tourism scenario description text is constructed by segmenting the documents describing the tourism scenario. The frequency $\tau_i = n_i/N$ of each text word $w_i$ is calculated. Here, n is the number of words $w_i$. N denotes the number of all words. Words with a frequency of $\tau_i \geq \varepsilon$ are selected as the tourism attribute's text description. The larger the word is, the higher the frequency of the word is. This word is added to the tourism scenario attribute semantic phrase as a semantic attribute expression of the tourism scenario. For example, 102 phrases describe the semantics of tourism scenario attributes in the SUN dataset. Furthermore, *n* = 102. Thus, it forms a 102-dimensional attribute space. A set of words can express the semantics of each attribute.

2.5.3. Tourism Scenario Attribute-Oriented Semantic Feature Relation Representation

(1)    Likelihood features of scenario attributes

The importance of the semantics of an attribute varies according to the topic in the scenario. The probability of an attribute belonging to this scenario is different. Here, the church scenario with visiting value is taken as an example. The prayer attribute of the scenario is a necessary attribute, and its likelihood probability is 1. The tourism-visit and time-honored attributes are also the main attributes, but they are not necessary. Therefore, the likelihood probability is *p* = 0.67. Aggregate and concrete attributes are non-essential

attributes, and their likelihood probability is $p = 0.33$. The value of each attribute is the logarithm of likelihood probabilities. Each attribute needs to be coded as a feature. The likelihood probability of the existence of scenario attributes is used to evaluate the coding, as shown in Equation (5).

$$P = [p^1 \; p^2 \quad p^n] \tag{5}$$

(2)    Semantic relationship features of scenario attributes

A group of text words expresses the semantics of each attribute. The text words need to be encoded since the text cannot directly participate in the calculation. The Skip-Gram model and word2vector method are used to encode text words. Each text word representing attributes can be expressed as a vector v through the word2vector model, as revealed in Equation (6).

$$v = word2vector(word) \tag{6}$$

Multiple text words describe the semantic expression of each attribute, and the number is uncertain. Therefore, in encoding the semantic features of an attribute, it is necessary to splice multiple word vectors to obtain the semantic feature code of the attribute. For feature alignment, the length of the attribute semantic feature vector of the maximum number of words is taken as the unified length. The attribute vector with insufficient length is filled with 0.

The semantics of the scenario is expressed by an attribute group G. An attribute group contains n attributes. Each attribute has k text word expressions. The attribute group of the scenario is a word vector matrix $M \in R^{N*D}$. Here, N is the number of attributes in the scenario attribute group. In $D = k \times d$, $k$ represents the maximum number of words in the attribute expression. $d$ denotes the dimension of the word vector of each word.

Due to the different topics of the scenario, the likelihood probability of each attribute occurrence is different, which is described by the likelihood vector *Likilyhood Vector (LV)* of the scenario attribute. Ideally, the occurrence probability of each attribute of the scenario is 1. The semantic relationship between the scenario attributes can be expressed as $SR = Sim$ $(M, M)$. As per the likelihood of the scenario attributes, the semantic feature is weighted. It can reflect the likelihood probability and attribute semantic information to the attributes of the scenario.

Suppose the semantic distance relationship between attributes $a_i$ and $a_j$ is $d_{i,j}$. $p_i$ and $p_j$ are the occurrence probability of attributes $a_i$ and $a_j$, respectively, which are expressed by likelihood: $p_i = Likilyhood(a_i)$ and $p_j = Likilyhood(a_j)$. When the two attributes co-exist, they have a closer semantic relationship expressed by Equation (7). The semantic relationship of scenario attributes is expressed by Equation (8). The vector of each attribute uses cosine distance to calculate attribute semantic distance.

$$r_{i,j} = (p_i + p_j)r_{i,j}, \; i,j \in N \tag{7}$$

$$d_{cos}(u,v) = 0.5 \times [1 - (u \times v)/(\|u\|\|v\|)], \; u,v \in M \tag{8}$$

In Equations (7) and (8), $r_{i,j}$ and $a_i$ are the semantic distance between attributes $a_i$ and $a_j$. $u$ and $v$ represent the vectors of attributes $a_i$ and $a_j$, respectively.

Then, the semantic relationship feature expression of the scenario attributes is obtained by Equation (9):

$$SR = Sim(PM,PM) = P \times Sim(M,M) \tag{9}$$

In Equation (9), *Sim* (*M, M*) is the semantic similarity of attributes. *M* denotes the semantic feature of attributes. *P* represents the likelihood vector of attributes.

It is well known that an image is a two-dimensional piece of data. Data can only be stored in one dimension in memory, and the digitization of video images mostly adopts quantitative methods. The following assumptions are made here. The tourist image data are arranged in the $M \times N$ dataset. Isometric sampling is used to obtain an approximately

continuous image $f(x, y)$. Then, the mathematical relationship shown in Equation (10) can be derived.

$$f(x, y) = \begin{bmatrix} f(0,0) & f(0,1) & \cdots & f(0,M-1) \\ f(1,0) & f(1,1) & \cdots & f(1,M-1) \\ \cdots & \cdots & \cdots & \cdots \\ f(N-1,0) & f(N-1,1) & \cdots & f(N-1,M-1) \end{bmatrix} \tag{10}$$

In Equation (10), the meaning of each element is a discrete variable.

The image digitization can be expressed in Equation (11).

$$\begin{cases} M = 2^m \\ N = 2^n \\ G = 2^k \end{cases} . \tag{11}$$

Here, the particle swarm algorithm is used to optimize the semantic expression algorithm of tourism scene attributes. In a Particle Swarm Optimization (PSO) system, a population of individuals (often called particles) moves through the search space. Each particle represents a potential solution to a particular optimization problem. The position of each particle in a population is influenced by the optimal position in its motion (individual experience) and the position of the optimal particle in its neighborhood (neighborhood experience). When the neighborhood of the particle is the entire particle population, the optimal position of the neighborhood corresponds to the global optimal particle. At this point, the algorithm is called the global PSO algorithm. Correspondingly, if a smaller neighborhood is utilized in the algorithm, it is usually called a local PSO algorithm. It is found that global PSO converges quickly, but it is easy to fall into local minima. Local PSOs can usually search for a better solution, but they are slower. In addition, in different optimization problems, a problem-related adaptation function is needed to evaluate the performance of each particle.

$f$ is used to represent the adaptation function. The individual optimal position $y_i$ of the particle $i$ can be adjusted according to the following equation.

$$y_i(t+1) = \begin{cases} y_i(t) & if \ f(x_i(t+1)) \geq f(y_i(t)) \\ x_i(t+1) & if \ f(x_i(t+1)) < f(y_i(t)) \end{cases} \tag{12}$$

In Equation (12), $x_i$ is the current position of particle $i$. $v_i$ is the current velocity of particle $i$. $y_i$ is the individual optimal position of particle $i$.

The particle neighborhood size is denoted as l and the particle population size as s. When l < s, the PSO algorithm at this time is the local PSO algorithm. When l = s, which means that the neighborhood of the particle is the entire population, the PSO algorithm at this time is the global version of the PSO algorithm. Then, the optimal position $\hat{y}$ of the group can be obtained according to the following equation.

$$\hat{y}(t) \in \{y_0, \ldots, y_s\} = min\{f(y_0), \ldots, f(y_s)\} \tag{13}$$

### 2.5.4. SASF Algorithm Flow

Based on the semantic relationship and likelihood features of the above tourism scenarios, the implementation process of the SASF algorithm is unfolded in Figure 3.

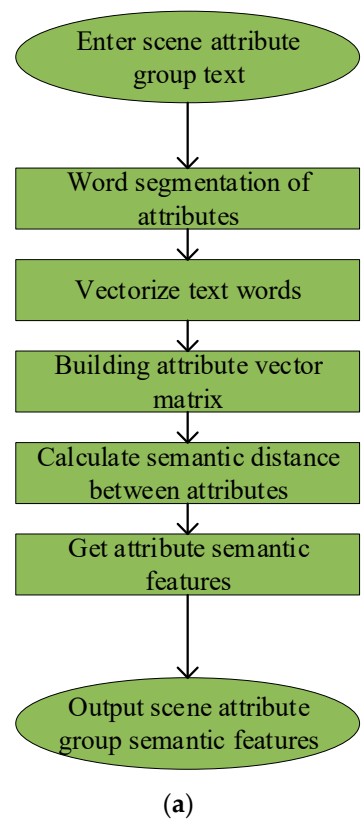

(**a**)

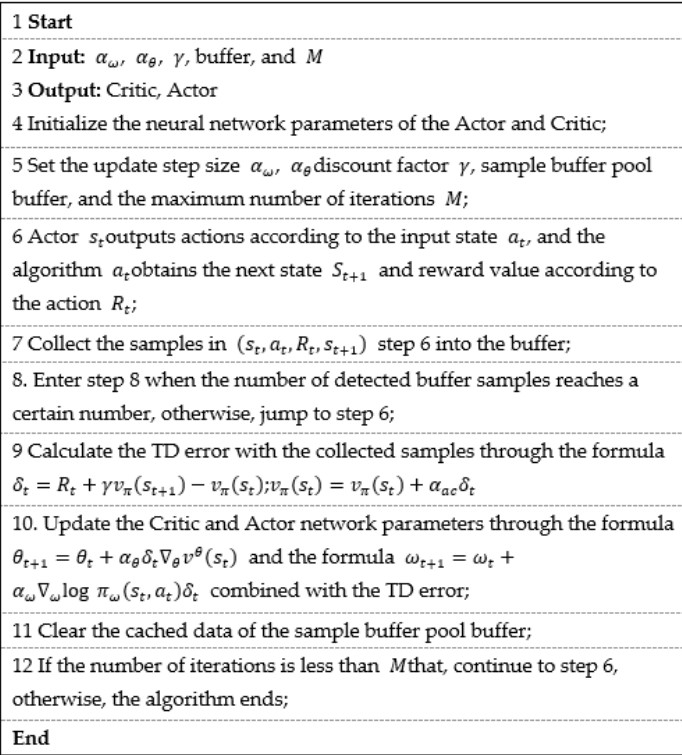

(**b**)

**Figure 3.** SASF algorithm flow. (**a**) process; (**b**) pseudocode.

### 2.5.5. Experimental Design

An experiment is designed to verify the proposed tourism scenario attribute-oriented semantic feature extraction method. The feature performance is tested and verified based

on multiple conventional classifiers, including K-means clustering (KMC), Support Vector Machine (SVM), and KNN classifiers. The verification experiment considers the semantic feature of scenario attributes and the likelihood feature LV of the scenario. Additionally, the experiment adopts scenario category recognition based on semantic features. It uses average accuracy to evaluate the effectiveness of features. The datasets and parameter settings are presented in Table 2.

**Table 2.** Experimental dataset and parameter selection.

| Dataset Name | Data Content | Attribute Scale | Scenario Type |
|---|---|---|---|
| **SUN** | Static image | 102 | 611 |
| **WWW Crowd Sub** | Dynamic character image | 94 | 5 |

### 3. Analysis of Rural Slow LCT Based on AI

The datasets required for the following research results are all from the literature on rural LCT collected in recent years.

#### 3.1. Analysis of Low-Carbon Slow Tourism

Figure 4 shows the environmental load brought by each tourist staying at a different time in the tourist destination.

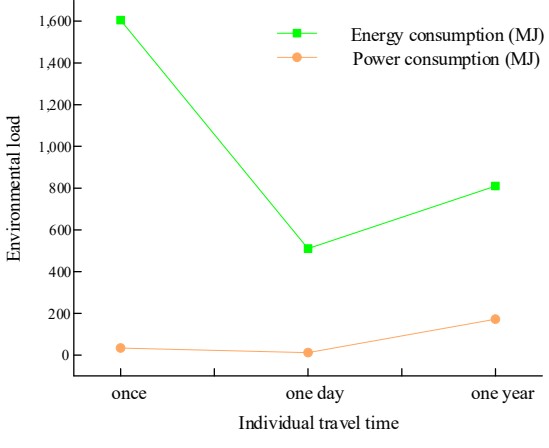

A

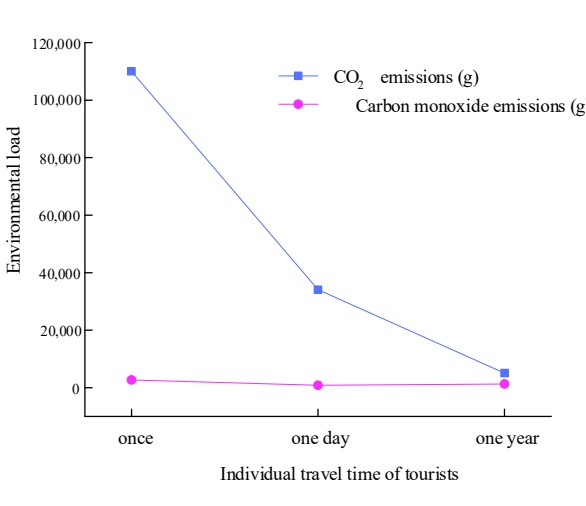

B

**Figure 4.** Environmental load brought by tourists. (**A**) Environmental load of energy. (**B**) Environmental load of air pollution.

The curve in Figure 4 shows the change degree of the environmental load on resources and environmental pollution when tourists travel over different time spans. Figure 4 reveals that when tourists travel to a specific place, they will bring different degrees of environmental load from the aspects of energy and air pollution, and this environmental load has a significant impact on water pollution and solid waste. Therefore, developing low-carbon slow tourism is the general trend in growing the modern tourism industry. Additionally, the realized degree of low-carbon slow tourism depends on the market, namely the support of tourists for low-carbon slow tourism [23].

Low-carbon tourists are those whose carbon emissions are zero or low, and they take the social responsibility to save energy in slow tourism, consume less energy, and bring no pollution during their journey [24].

### 3.2. Analysis of Rural Slow LCT

China is a country with a vast territory, and there is much rural and natural scenery with folk customs. These beautiful sceneries gradually receive people's attention with the development of society, economy, and information. People watch this kind of natural scenery to precipitate their glitz and improve themselves in the rural landscape with different purposes. Regarding this, the rural slow tourism industry in China has begun to flourish and forms a unique tourism model in the tourism industry [25].

Figure 5 illustrates the two spatial elements in the formation of rural slow LCT.

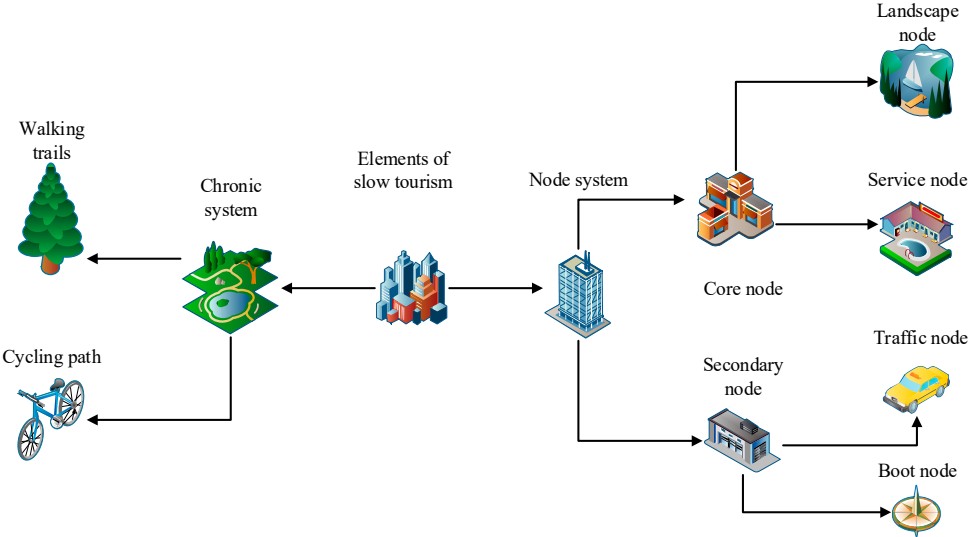

**Figure 5.** Elements of slow tourism.

From Figure 5, the core of the slow tourism system is the node system, which includes the core node and the secondary node. The core node is the basic service node related to the sightseeing and tourism elements in the tourist destination, which ensures that constructing the main scenic spots from the relevant basic services to the tourist destination can meet the needs of tourists. Among them, the landscape node mobilizes the participation, experience, and culture of tourists. Therefore, tourists can have a deep experience of local culture and integrate into the local cultural spirit of life. The secondary nodes improve the public facilities of the tourist destinations so that tourists have a better travel experience. As the track of the whole tour, the slow travel system is responsible for connecting the whole paralysis tourism system. In the design, the elements of travel should be considered. In the construction of slow tourism, the existence of road basic attributes should also conform to the humanized design, ensuring the experience of tourists in the process of slow tourism [26].

Figure 6 shows the factors that are included in the development of slow rural tourism.

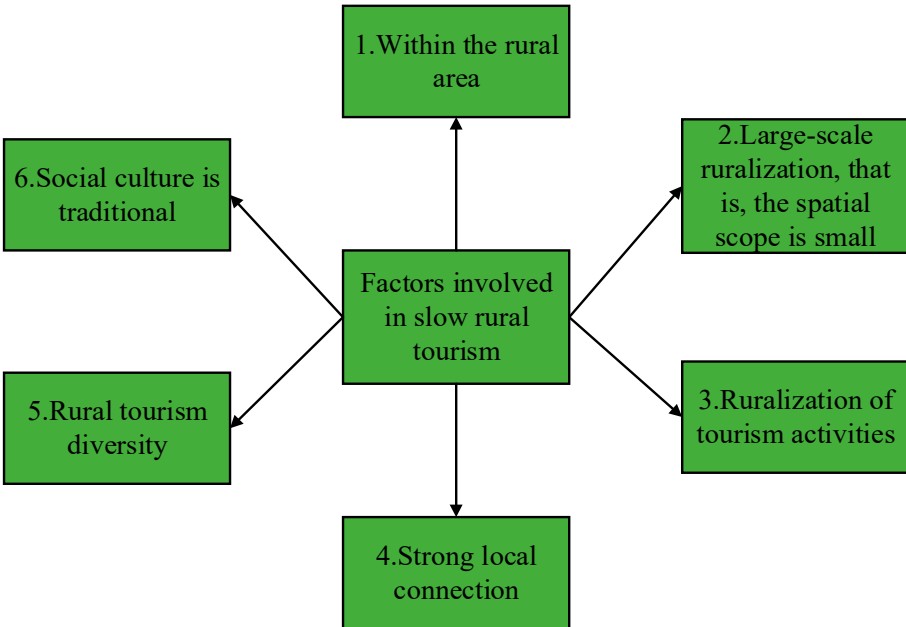

**Figure 6.** Factors of slow rural tourism.

From Figure 6, there are several important factors contained in slow rural tourism after the relevant literature is reviewed. Rural slow tourism is explained in many ways, but its explanations are mainly elaborated from the subject, the object, and the destination. In slow rural tourism, many urban residents are attracted. Rural space, natural features, and humanistic culture are the factors that attract tourists. In tourism, tourists can enjoy sightseeing and experience nature. With the development of slow rural tourism in China, more and more people are involved in rural construction. Based on this, a new concept of slow rural tourism is generated. In the new concept of original rural slow tourism, there are such modes as wandering, wilderness, and poetic dwelling. The core of these different forms is rurality, which is the unique style and local customs of rural areas [27].

Figure 7 plots the number of key rural slow tourism villages in some regions of China.

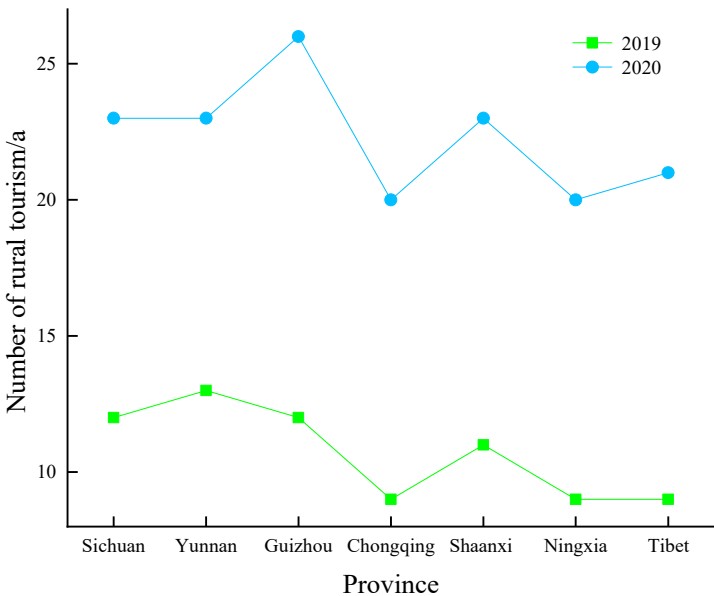

**Figure 7.** The number of key rural slow tourism villages in some regions.

Figure 7 shows the number of key rural slow tourism villages in some provinces and cities nationwide. The number of slow rural tourism sites in Guizhou has increased the most in these provinces and cities, while the number of slow rural tourism sites in Yunnan has increased a little. Combined with its geographical location and humanistic characteristics, it is found that the multi-ethnic settlement areas are rural slow tourism villages based on their administrative villages or ethnic natural villages, and they make outstanding contributions to the slow tourism development and economic development of the region [28].

Figure 8 displays the proportion of carbon emissions in each part of the slow tourism industry.

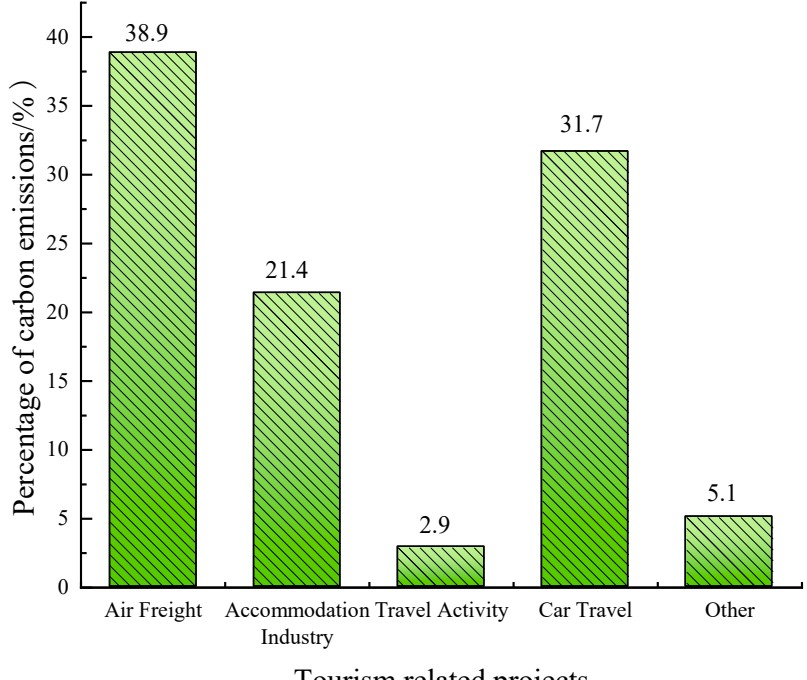

**Figure 8.** The proportion of carbon dioxide emissions in the tourism industry.

From Figure 8, the carbon dioxide emissions of one-day tourism projects account for less than 10% of the total in the whole tourism industry. From the proportion analysis, the largest proportion comes from air transport of more than 40%. These data truly reflect the general level of the carbon dioxide emissions of the tourism industry, which is a relatively reliable quantitative analysis result at present. With the continuous and rapid development of the tourism industry, many problems appear due to carbon emissions, and they include climate warming and the shortening of winter snowfall time, which make all kinds of snow-related projects rely on artificial snow technology. Therefore, the carbon emissions of the tourism industry need to be paid attention to, and the reduction in carbon dioxide emissions through LCT is also a necessary trend [29].

Figure 9 shows the total and per capita carbon emissions in western China.

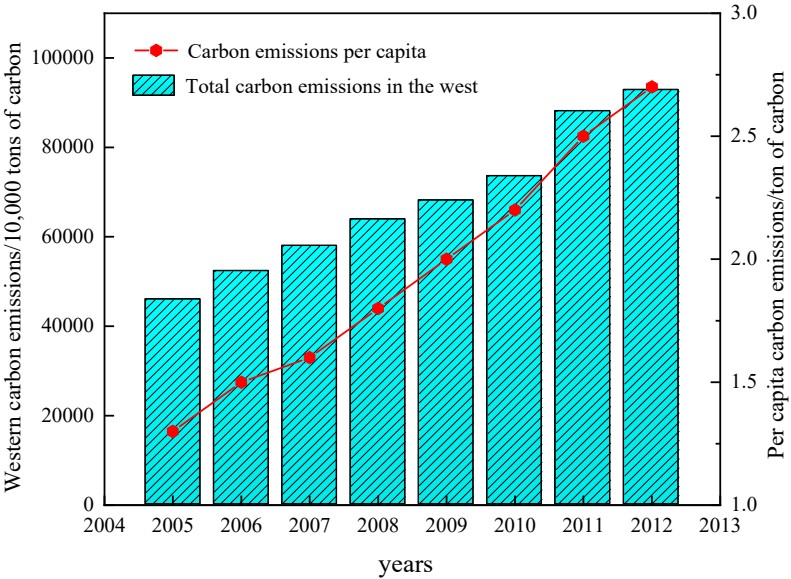

**Figure 9.** Carbon emissions and per capita carbon emissions in western China.

From Figure 9, the total amount of carbon emissions and per capita carbon emissions in the western region of China increases year by year, indicating that the total amount of carbon emissions in China is also increasing year by year with the development and progress of society, and the fluctuation of carbon emissions is changing. Many carbon emissions make the damage to the environment more serious. Therefore, carbon emissions should be considered carefully in developing tourism and pursuing more rural slow LCT [30].

At present, the development of rural slow LCT in China is still at a low level, which causes more negative effects, including environmental pollution and resource destruction. At the beginning of constructing slow rural tourism, the noise brought by tourists breaks the tranquility of the countryside. The exhaust gases brought by self-driving tours pollute the overall environment of the countryside. Untreated sewage and discarded garbage also pollute the rural environment [31]. In addition, at the beginning of rural slow tourism, many enterprises only pursue economic benefits and over-develop rural slow tourism resources. The arrival of many tourists also destroys the environmental balance, destroying resources.

A low-carbon life in both production and life should be maintained to form low-carbon rural slow tourism, including low-carbon energy, low-carbon awareness, and rational use of land, which will be transmitted into all aspects of rural slow tourism. Various ways should be used to develop the education of rural slow LCT, contributing to the development of low-carbon rural areas [32].

*3.3. Analysis of Rural Slow LCT Based on AI*

Figure 10 shows an important reason for developing rural slow LCT in China or the feasibility of developing rural slow LCT in China.

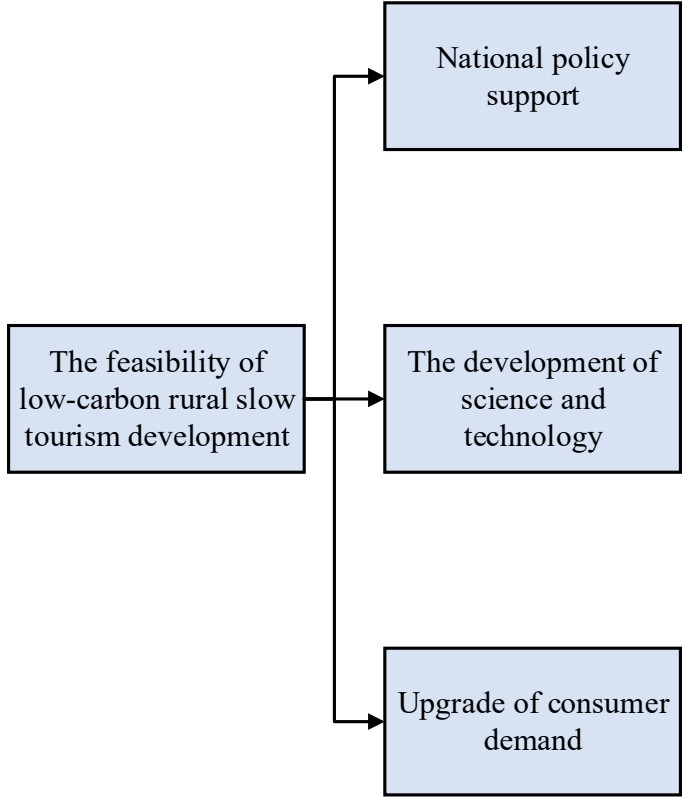

**Figure 10.** Feasibility of developing rural slow LCT.

Figure 10 presents the feasibility of the development of rural slow LCT in China. Through the above analysis, it is found that there is still much room for improving slow rural tourism in China. In recent years, with the improvement in science and technology and people's living standards, rural slow LCT has become an inevitable trend [33].

Figure 11 shows the principles of rural slow LCT planning based on AI.

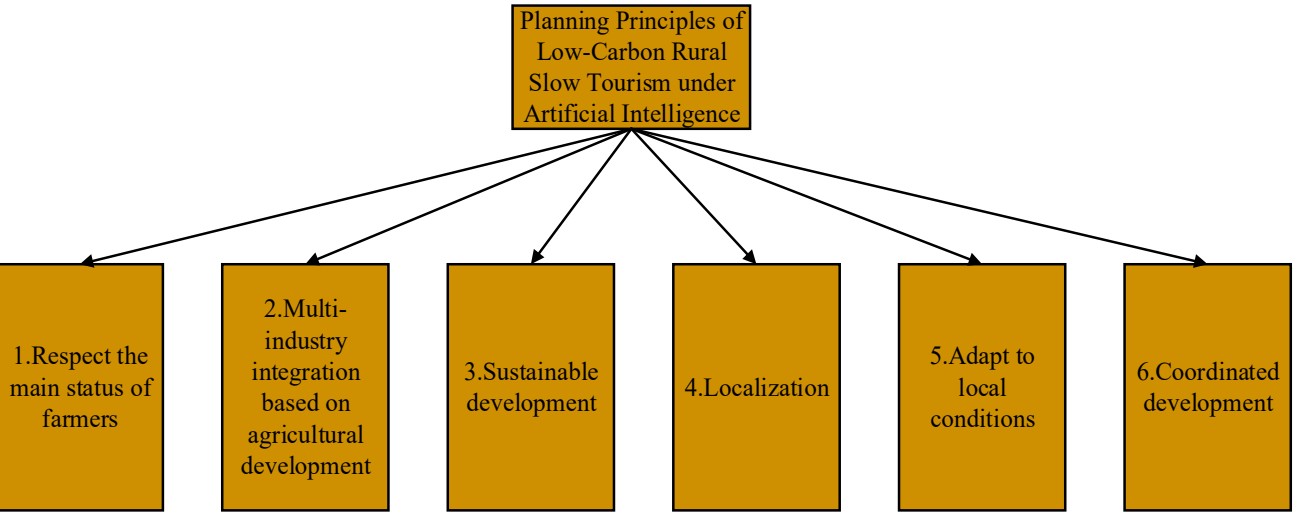

**Figure 11.** Principles of rural slow LCT planning.

Figure 11 depicts the principles in the planning of rural slow LCT. These principles can help to obtain the most accurate data under the application of AI, and the most suitable example of rural slow LCT is found according to the analysis of these data, providing the optimal scheme for the area of rural slow tourism [34].

Figure 12 illustrates the countermeasures for rural slow LCT planning under AI.

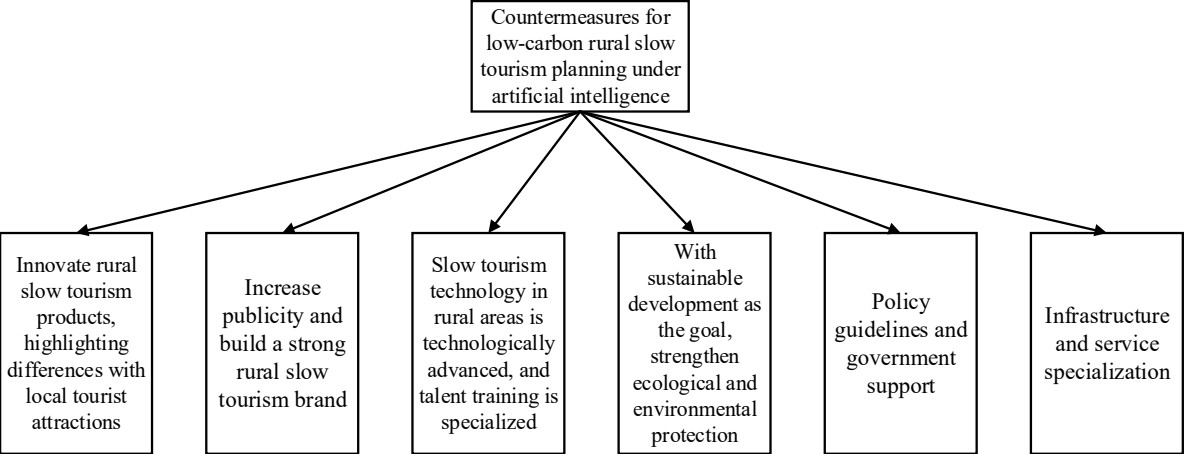

**Figure 12.** Countermeasures for rural slow LCT planning under AI.

Figure 12 shows the countermeasures for rural slow LCT planning under AI. Appropriate countermeasures can help tourism development. Through the analysis and research on modern people's tourism intention and rural slow LCT data based on AI, the most superior countermeasures can be summarized. These countermeasures can help slow tourism in rural areas to embark on the right track faster and promote the development of rural slow LCT [35].

The planning of rural slow LCT based on AI can meet and complete the requirements of the sustainable development of rural tourism. These planning principles and countermeasures can make the farmers of rural slow LCT spend the least time finding out about rural slow LCT and know about the situation of rural slow LCT in the fastest way. In this case, a suitable planning route for developing slow rural tourism is designed so that rural slow LCT can benefit people and provide high-quality services and the best experience of rural slow LCT for tourists [36].

Moreover, geographic information about rural areas can be collected and analyzed on the network platform because of the development of AI. Combined with the results of field visits, the geographic information of rural areas can be visualized in the virtual world. This is not only a summary of the geographical information of the rural slow tourism area but also a more advanced and complete technical means to see the specific situation of the tourism area for those who are interested in or for whom it is temporarily inconvenient to come [37]. The information dissemination speed under AI is very fast. After the corresponding rural slow tourism data are updated and searched, they can recommend the optimal rural area that may be selected for tourists according to the data analysis of equipment browsing records [38].

Figure 13 shows the results of the survey on land use types in different villages and towns.

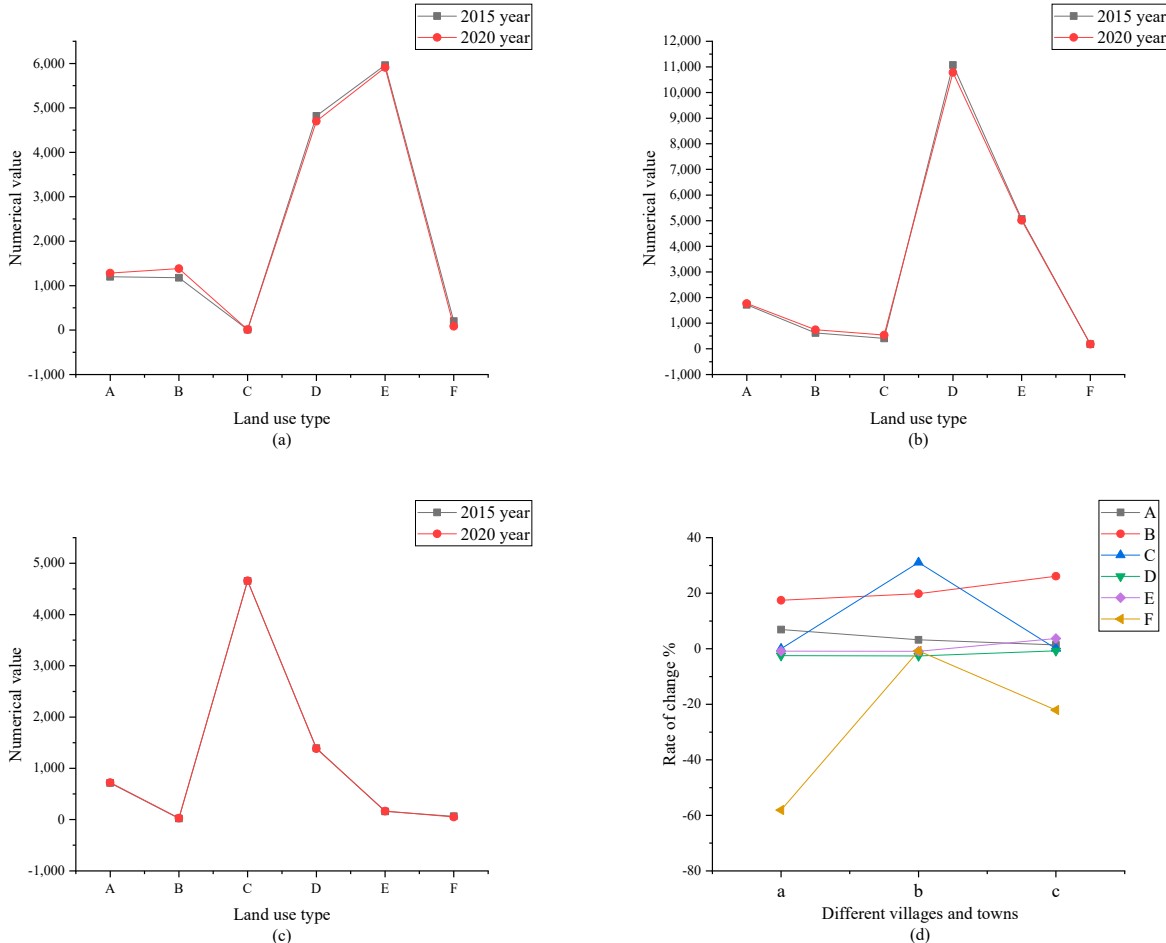

**Figure 13.** Survey of land use types in different villages and towns. (A: rural settlements; B: industrial and mining construction land; C: woodland; D: agricultural land; E: water bodies; F: other uses; (**a**) ancient town A; (**b**) ancient town B; (**c**) ancient town C; (**d**) rate of change in land use type).

It is found that the area change in ancient town A is more obvious than that of the other two ancient towns by comparing the land area changes in ancient town A, ancient town B, and ancient town C, and the change rate is 6.98%. Although the area change rate of the ancient town is 19.79%, the actual change area is not as much as that of ancient town A. This is because the area of ancient town B is relatively small, resulting in a high rate of change. In terms of agricultural land, the land change rate in ancient town C is the smallest. The rates of change of ancient town A and ancient town B are 2.51% and 2.62%, respectively. The reason for this situation is that ancient town C is located on the Loess Plateau, and its water resources are relatively scarce.

### 3.4. Tourism Scenario Recognition Effect Based on the SASF Algorithm

3.4.1. Experimental Results and Analysis of the SASF Algorithm on the SUN Dataset

The proposed SASF algorithm is tested on the SUN scenario attribute dataset. The results of scenario classification based on scenario attribute features using multiple classifiers are compared in Figure 14.

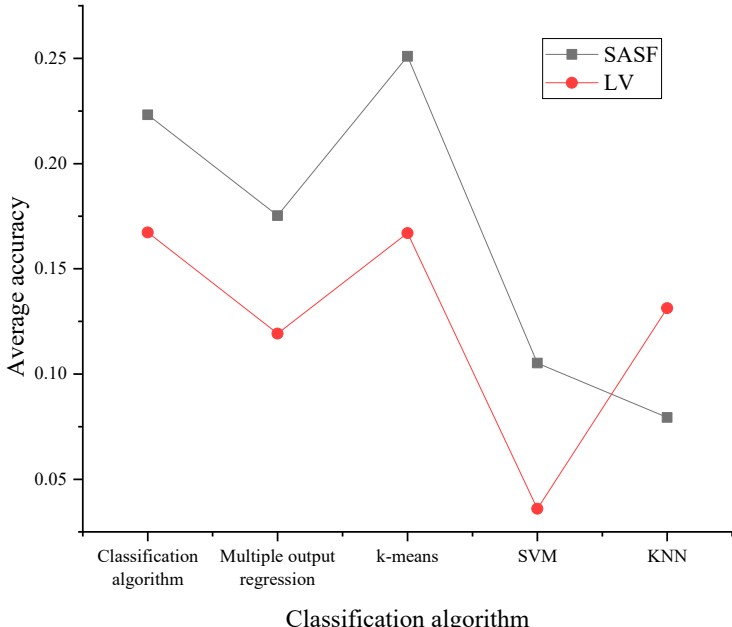

**Figure 14.** Experimental results of tourism scenario recognition based on the SASF algorithm on the SUN dataset.

Figure 14 shows that in the KMC model, the accuracy of scenario classification based on SASF is improved by 5.26% more than that of the attribute likelihood vector. On the SVM classifier, the scenario classification accuracy based on SASF is 192% higher than that based on the attribute likelihood vector feature. The reason for the fluctuation in recognition accuracy is the complexity of the recognized picture.

3.4.2. Experimental Results and Analysis of the SASF Algorithm on the SUN Dataset

The proposed SASF algorithm is tested on the WWW Crown Sub scenario attribute dataset. The results of scenario classification based on scenario attribute features using multiple classifiers are analyzed in Figure 15.

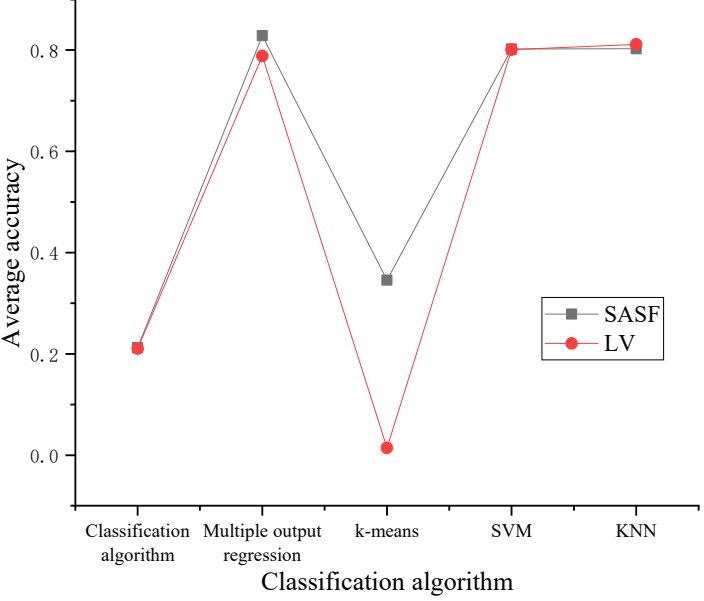

**Figure 15.** Experimental results on the WWW Crown Sub dataset of the tourism scenario recognition method based on the SASF algorithm.

Figure 15 indicates that the SASF algorithm presents better accuracy in scenario recognition than using only likelihood vector features in different conventional classifiers. Overall, the recognition accuracy is excellent.

The SASF algorithm proposed here is tested on the CoCo dataset. Figure 16 compares the scene classification results based on scene attribute characteristics using multiple classifiers.

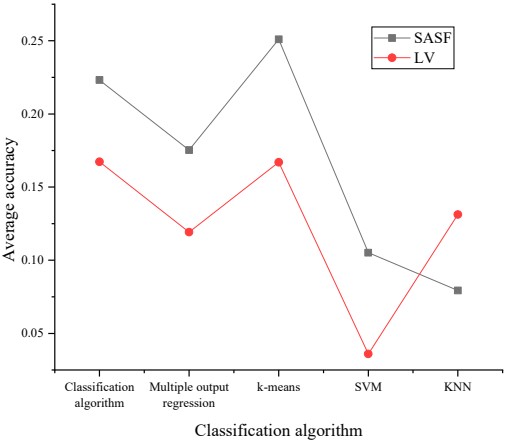

**Figure 16.** Experimental results of tourism scenario recognition based on the SASF algorithm on the CoCo dataset.

Figure 16 shows that on the CoCo dataset, the accuracy of SASF-based scenario classification is 5.26% higher than that of attribute likelihood vectors. On SVM classifiers, the scene classification accuracy based on SASF is 19.2% higher than that based on attribute likelihood vector features.

3.4.3. Discussion

Based on the previous research, it is found that the scale of public space for rural slow tourism should be small and precise. The creation of rural slow tourism space requires a long process. Intensive space creation is needed to minimize the impact on rural space for the space to be sustainable. Moreover, in the process of space creation, attention should be paid to the grasp of spatial scale to avoid the damage to the countryside caused by overexploitation of the countryside.

The theoretical significance is as follows: (1) The first is to help establish the concept of slow tourism and clarify the characteristics of slow tourism. Through the study and research of a large amount of Chinese and foreign literature, the development process of the concept of "slow tourism" and its related similar concepts such as slow food, slow city, and slow life is sorted out. The concepts of Chinese and foreign scholars and slow tourism advocates are compared and studied. The concept of slow tourism is proposed, and the characteristics of slow tourism impact and tourist behavior are elaborated. (2) The second is to improve the research system of rural tourism. The rural area is developing at a high speed. Today, "rural tourism" has become a high-frequency word in society. How to not only realize the protection of rural areas but also promote the development of the rural economy requires more theoretical methods and path exploration. The rural slow tourism discussed here is a supplement to the research system of rural tourism. Moreover, it is an exploration and attempt to integrate the concept of slow tourism into rural tourism.

The practical significance of this study: (1) The first is to help rural revitalization. Promoting urban and rural development and rural tourism is the core of targeted poverty alleviation through tourism. It has played a huge role in solving the three rural problems, expanding the value chain of the agricultural industrial chain, helping poverty alleviation, and coordinating the construction of urban and rural areas. Rural slow tourism can not only meet the needs of people yearning for a "better life" but also solve the current "unbalanced

and insufficient development" relationship between urban and rural areas. It has important enlightenment significance for how China's vast rural areas can identify development goals in the context of urbanization and marketization. Moreover, as a development model truly suitable for rural areas, rural tourism is an important starting point for the rural revitalization strategy and the key to achieving sustainable rural development. (2) The second is to improve the spatial connotation, nourish the culture, and cultivate the cultural consciousness of the villagers. Merely developing villages without benefiting villagers will lead to a psychological imbalance among villagers. Developing only the villagers without developing the village will lead to the decay of the village. Villagers need to identify with the places where they live to achieve a balanced development of the countryside. Villagers need to spontaneously understand the cultural connotation of the place they live in and understand its formation process and development characteristics. The enthusiasm of the broadest masses of people can be aroused. In addition, their creativity and productivity will be unleashed. The rural slow tourism space can carry the rural cultural image and landscape intention. The creation of rural slow tourism space is the reproduction and expression of rural cultural and landscape intention. It can provide a space and atmosphere for villagers to have cultural exchanges in the village. (3) Based on the in-depth analysis of rural slow tourism space, the strategy of rural slow tourism space construction is proposed. Through the collation and summary of the literature, the research status of rural tourism in slow tourism is analyzed. Based on the full understanding of relevant concepts, the concept of rural slow tourism is pioneered. The strategy of rural slow tourism space creation is proposed based on the results of theoretical research and the study of specific cases.

## 4. Conclusions

Based on AI, the related theoretical concepts of rural slow LCT are studied, and the algorithm following DL is put forward, the theories and survey data results of LCT, slow tourism, and rural tourism are analyzed, and the geographic information visualization and sustainability of rural slow LCT from AI are explored. According to the CMR technology, a tourism scenario recognition method based on SASF is constructed. Through the analysis and research, it is found that developing LCT is the inevitable trend of tourism development. The combination of slow tourism and rural slow tourism is the way of tourism pursued by people under the pressure of modern life. The geographic information visualization of rural slow LCT under AI enables people to see the local folk features and characteristic culture through rural slow tourism so that people have a preliminary impression of the projects of tourism destinations. The visual information extracted by AI can arouse people's interest in rural culture. Through this information, tourists can also know about and contact the environment faced by slow rural tourism earlier, which helps to increase tourists' sense of identity and familiarity with local culture in travel. On the other hand, experiments prove that the SASF algorithm presents higher accuracy for text classification in tourism scenario attributes. The innovation is that it discusses and analyzes the geographical information visualization and sustainable development of rural slow LCT. Moreover, it proposes a tourism scenario recognition method using CMR technology. The proposed method has a good recognition effect and saves tourists' preparation time by improving the recognition efficiency of tourism scenes, achieving the LCT goal. The concept of rural slow tourism is summarized using the literature review and comprehensive analysis. Meanwhile, its composition is analyzed spatially. A lot of time has been spent on summarizing and classifying domestic and international research reviews, but no more in-depth quantitative studies and analyses of tourist activities have been conducted on the cases. Moreover, there are inevitably omissions due to the limited time for case studies. In the future, the duration of research can be extended, preferably including holidays and non-weekends. The content of research for tourists can be increased. China's rural areas have the creative spirit of hundreds of millions of farmers. It is supported by strong policy and economic strength. It carries a long history of agricultural civilization and has a strong market demand. As a new tourism concept, rural slow tourism has great research value in exploring the creation

strategy of rural slow tourism space based on the full excavation of rural resources and rural protection. How to better apply rural slow tourism to rural areas and how to create a spatial atmosphere of rural slow tourism from the spatial level requires further empirical research and theoretical exploration.

**Author Contributions:** Conceptualization, G.J. and W.G.; methodology, M.X.; software, M.T.; validation, Z.L. and M.T.; formal analysis, G.J.; investigation, M.X.; data curation, W.G.; writing—original draft preparation, G.J.; writing—review and editing, M.X.; visualization, Z.L.; supervision, M.X.; All authors have read and agreed to the published version of the manuscript.

**Funding:** This research received no external funding.

**Institutional Review Board Statement:** Not applicable for studies not involving humans or animals.

**Informed Consent Statement:** Not applicable for studies not involving humans.

**Data Availability Statement:** The raw data supporting the conclusions of this article will be made available by the authors without undue reservation.

**Conflicts of Interest:** The authors declare no conflict of interest.

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
