# Peer review of "Geographic Information Visualization and Sustainable Development of Low-Carbon Rural Slow Tourism under Artificial Intelligence"

_sustainability, doi:10.3390/su15043846_

Round 1
Reviewer 1 Report
This article should be more coherent to be published in this journal. Therefore, the following corrections can help the quality of this article
1- The abstract should be modified (introduction, method, results)
2- The purpose of the research should be stated more clearly in the abstract.
3- The introduction needs to be rewritten. In the introduction, it should be written in relation to research variables and theoretical gaps.
The research literature was not found in this article.
While many studies have been conducted in relation to the subject.
4- Bring the description of the model in the appendix, and focus more on the literature.
5- The discussion was not observed in the article. Add a discussion section to the article that includes theoretical and practical implications.
6- Give more explanations about the limitations of research and future research
Author Response
This article should be more coherent to be published in this journal. Therefore, the following corrections can help the quality of this article
1- The abstract should be modified (introduction, method, results)
Reply: Thank you for your review comments. Your review comments have provided great help for us to revise the article. During the revision of the article, we have carefully revised the abstract of the article and added the background, methods and results of the article's research. The research results of this article are that the number of slow tourists in rural areas in Guizhou has increased the most over time, while the number of slow tourists in rural areas in Yunnan has increased slightly. In the K-means clustering model, the accuracy of scene classification based on the semantic features of scene attributes is 5.26% higher than that of the attribute likelihood vector. On the SVM classifier, the accuracy of scene classification based on the semantic feature of scene attributes is 19.2% higher than that based on the feature of attribute likelihood vector.
2- The purpose of the research should be stated more clearly in the abstract.
Reply: Thank you for your comments, which is very helpful for us to revise the article. When the article was revised, we added the main purpose of this study in the abstract and introduction of the article. The main purpose of this study includes the following two aspects: 1. Interpret the connotation of rural slow travel and initially establish the research framework of rural slow travel space. 2. Interpret the requirements of rural revitalization strategy for rural tourism, and make clear the starting point of spatial research.
3- The introduction needs to be rewritten. In the introduction, it should be written in relation to research variables and theoretical gaps.
The research literature was not found in this article.
While many studies have been conducted in relation to the subject.
Reply: Thank you for your opinion. We rewrote the introduction of the article in connection with the blank of the research theory. The blank of the research theory is that the existing literature has studied low-carbon tourism, but it is mainly aimed at the design of green tourism schemes, and the research object is non-rural tourist destinations. There is little research on rural slow tourism and low-carbon tourism. Based on this, starting from the rural tourism scene, this study explores new ways to realize low-carbon tourism by using various machine learning methods, and modifies the reference list in the article according to the revised article.
4- Bring the description of the model in the appendix, and focus more on the literature.
Reply: Thank you for your comments. During the revision of the article, we have carefully revised the reference appendix in the article and added more references on model research and improvement.
5- The discussion was not observed in the article. Add a discussion section to the article that includes theoretical and practical implications.
Reply: Thank you for your comments. We supplemented the discussion of the results of the article in the "3.4.3" section of the article. This article believes that the scale of public space showing slow rural tourism should be small and precise. The sign system of slow tourism is the key part of creating slow tourism space.
6- Give more explanations about the limitations of research and future research
Reply: Thank you for your review and comments. During the revision of the article, we added the limitations of the research and the outlook for future research in section 4 of the article. The limitations of the research are reflected in the fact that the author spent a lot of time summarizing and classifying domestic and foreign research reviews, and there is no need to carry out more in-depth quantitative research and analysis of tourist activities on the case. There are certain omissions in the research.
Reviewer 2 Report
What is the novelty of this work? I am unable to find what is the significance of this work.
The literature review is missing. It is not clear why this is required. i suggest to make a table with pros and cons to clarify what exactly are the missing sets.
The setting the scene is missing. The important details that how this method is implemented is missing.
The framework of the system is missing. What is the mathematics behind this?
The pseudo code is missing.
Results are inconclusive. Better results are required to see the efficacy.
Author Response
What is the novelty of this work? I am unable to find what is the significance of this work.
The literature review is missing. It is not clear why this is required. i suggest to make a table with pros and cons to clarify what exactly are the missing sets.
Reply: Thank you for your comments. When the article was revised, we rewrote the introduction of the article in view of the blank of research theory. The blank of research theory is that the existing literature has studied low-carbon tourism, but it is mainly aimed at the design of green tourism scheme, and the research object is non-rural tourism destinations. There is little research on rural slow tourism and low-carbon tourism. Based on this, this study explores new ways to realize low-carbon tourism by using various machine learning methods from rural tourism scenes. In the introduction of the article, we added references 1) MM Nasrallah Sustainability 13 (9), 4716, 2) Chen Y, Chen Y, Chen K, Liu M Research progress and hot spot analysis of residential carbon emissions based on CiteSpace software. International journal of environmental research and public health. January 2023; 20(3):1706., 3) Dai X, Chen Y, Zhang C, He Y, Li J. Technological revolution in agriculture: the green development of agriculture in China driven by DIT. Agriculture. January 2023; 13 (1): 199. The innovation of this paper lies in the pioneering introduction of the concept of rural slow tourism on the basis of full understanding of related concepts, and combining the results of theoretical research with the study of specific cases, this paper puts forward the strategy of building rural slow tourism space.
The setting the scene is missing. The important details that how this method is implemented is missing.
Reply: Thank you for your review work and comments. Your review comments have provided great help for us to modify the article. The details of SASF algorithm are added in the section "2.5.3" of the article.
The framework of the system is missing. What is the mathematics behind this?
Reply: Thank you for your comments. When the article was revised, we added the mathematical basic principle of semantic expression of tourism scene attributes in the "2.5.3" section of the article.
The pseudo code is missing.
Reply: Thank you for your comments. When the article was revised, the pseudo-code of the algorithm was added in Figure 3 of the "2.5.4" section of the article.
Results are inconclusive. Better results are required to see the efficacy.
Reply: Thank you for your review and comments. Your comments are of great help to us in revising the article. We supplemented the test results of SASF algorithm proposed in this paper on CoCo data set in the "3.4.2" section.
Reviewer 3 Report
The authors analyze and discuss the geographic information visualization and sustainable development of rural slow and Low-Carbon Tourism. The paper is technically sound and overall a good read. Some minor issues should be corrected in the manuscript to improve it further.
1. In the Introduction, more emphasis should be laid on the use of Deep-Learning-based Computer Vision. This can help to provide the necessary background for carrying out this work. The outline and contribution of the study should also be mentioned.
2. In the related work section, highlight the issues or challenges reported in previous works. This would help the reader to understand the limitation of existing works, you may need to expand the literature by including the following references: 1) Sustainable virtual reality patient rehabilitation systems with IoT sensors using virtual smart cities MM Nasralla Sustainability 13 (9), 4716, 2) Chen Y, Chen Y, Chen K, Liu M. Research Progress and Hotspot Analysis of Residential Carbon Emissions Based on CiteSpace Software. International Journal of Environmental Research and Public Health. 2023 Jan;20(3):1706., and 3) Dai X, Chen Y, Zhang C, He Y, Li J. Technological Revolution in the Field: Green Development of Chinese Agriculture Driven by Digital Information Technology (DIT). Agriculture. 2023 Jan;13(1):199.
3. The authors should provide more details on the data flow model.
4. I am satisfied with the results provided. There should be more datasets running to obtain more stable graphs. Maybe these results can be presented in tables.
5. Please provide a clear justification of why the results are going up and down
6. some typographical and grammatical mistakes should be checked and corrected.
Author Response
1. In the Introduction, more emphasis should be laid on the use of Deep-Learning-based Computer Vision. This can help to provide the necessary background for carrying out this work. The outline and contribution of the study should also be mentioned.
Reply: Thank you for your review and comments. During the revision of the article, we supplemented the application of deep learning and computer vision technology in rural tourism in the introduction of the article, and supplemented the outline and contribution of the article in the last natural paragraph of the introduction. The contribution of this article is to analyze the current research situation of rural tourism and slow tourism through combing and summarizing the literature. On the basis of fully understanding the relevant concepts, the concept of rural slow tourism was introduced creatively, and the strategy of building rural slow tourism space was put forward based on the results of theoretical research and the study of specific cases.
2. In the related work section, highlight the issues or challenges reported in previous works. This would help the reader to understand the limitation of existing works, you may need to expand the literature by including the following references: 1) Sustainable virtual reality patient rehabilitation systems with IoT sensors using virtual smart cities MM Nasralla Sustainability 13 (9), 4716, 2) Chen Y, Chen Y, Chen K, Liu M. Research Progress and Hotspot Analysis of Residential Carbon Emissions Based on CiteSpace Software. International Journal of Environmental Research and Public Health. 2023 Jan;20(3):1706., and 3) Dai X, Chen Y, Zhang C, He Y, Li J. Technological Revolution in the Field: Green Development of Chinese Agriculture Driven by Digital Information Technology (DIT). Agriculture. 2023 Jan;13(1):199.
Reply: Thank you for your review comments. Your review comments have provided us with a lot of help in revising the article. In the introduction of the article, we added references 1) Sustainable virtual reality patient rehabilitation systems with IoT sensors using virtual smart cities MM Nasralla Sustainability 13 (9), 4716, 2) Chen Y, Chen Y, Chen K, Liu M Research Progress and Hotspot Analysis of Residential Carbon Emissions Based on CiteSpace Software. International Journal of Environmental Research and Public Health. 2023 Jan; 20(3):1706., and 3) Dai X, Chen Y, Zhang C, He Y, Li J. Technological Revolution in the Field: Green Development of Chinese Agriculture Driven by Digital Information Technology (DIT). Agriculture. 2023 Jan; 13 (1): 199.
3. The authors should provide more details on the data flow model.
Reply: Thank you for your review work and comments. When the article was revised, we added more details of the data flow model in the "2.5.1" section of the article. Among them, dynamic pure data flow is the core of many graphical programming platforms, and its natural properties cannot be well combined with the event-driven model of the operating system, which leads to two obvious shortcomings: low running efficiency and high CPU utilization. This paper proposes an event-triggered concurrent data flow model.
4. I am satisfied with the results provided. There should be more datasets running to obtain more stable graphs. Maybe these results can be presented in tables.
Reply: Thank you for your review and comments. Your comments are of great help to us in revising the article. When the article was revised, we supplemented the test results of SASF algorithm proposed in this paper on CoCo data set in the article "3.4.2".
5. Please provide a clear justification of why the results are going up and down
Reply: Thank you for your comments. The reason why the results of the original Figure 13 (now Figure 14) going up and down is that the complexity of the algorithm to identify pictures is different.
6. some typographical and grammatical mistakes should be checked and corrected.
Reply: Thank you for your review comments and work. When the article was revised, we revised the typographical errors and grammatical errors in the article.
Round 2
Reviewer 1 Report
The article has been improved, therefore, I recommend this article for publication in this journal
Reviewer 2 Report
The scope of article is still not clear. Again, what is the novelty of this work? Authors claim to use AI but I don't see any AI based scheme or framework?
If AI method is used, add the following:
1. Schematic of the architecture. Is it CNN, SVM or what? because in abstract you claim it is SVM but in the pseudocode you say it is neural network? Which one is it? How it is linked to actor critic here? actor critic is generally used in reinforcement learning scenario. what are the learning parameters that you are using for classifier?
2. Training and validation error curves.
3. Confusion matrix
4. Comparison of complexity of the proposed AI
Why PSO was used? It is not novel and doesn't add any novelty to the already defined work. The use of PSO should be justified. Show the result with and without PSO. How does it increases the complexity of the system?
Is it experimental or simulation? Again, setting the scene is missing, it means the simulation or experimental test bench, the settings of the simulations, parameters should be shown in a Table.
Reviewer 3 Report
Thank you.